# Teaming LLMs to Detect and Mitigate Hallucinations

**Demian Till**[1]* **John Smeaton**[1] **Peter Haubrick**[1] **Gouse Saheb**[1]
**Florian Graef**[1] **David Berman**[2]
[1] Cambridge Consultants    [2] Queen Mary University of London

## Abstract

Recent work has demonstrated state-of-the-art results in large language model (LLM) hallucination detection and mitigation through consistency-based approaches which involve aggregating multiple responses sampled from a single LLM for a given prompt. These approaches help offset limitations stemming from the imperfect data on which LLMs are trained, which includes biases and under-representation of information required at deployment time among other limitations which can lead to hallucinations. We show that extending these single-model consistency methods to combine responses from multiple LLMs with different training data, training schemes and model architectures can result in substantial further improvements in hallucination detection and mitigation capabilities beyond their single-model consistency counterparts. We evaluate this *consortium consistency* approach across many model teams from a pool of 15 LLMs and explore under what conditions it is beneficial to team together different LLMs in this manner. Further, we show that these performance improvements often come with reduced inference costs, offsetting a significant drawback with single-model consistency methods.

## 1 Introduction

A well-known, major limitation of current LLMs is their propensity to hallucinate, producing plausible but factually-incorrect responses. Quality of pre-training data and instruction fine-tuning data plays a key role in hallucination behavior. When information relevant to deployment-time performance is under-represented or misrepresented in the pre-training corpus, the model is less likely to be able to provide accurate responses [1, 2]. Moreover, instruction fine-tuning can incentivize models to make educated guesses in the absence of reliable knowledge on a given topic [3]. During instruction fine-tuning it is relatively expensive to determine what a model genuinely does not know and to include fine-tuning examples which encourage it to admit when it does not know something. Such examples are therefore likely to be underrepresented in typical fine-tuning data, thereby providing insufficient counterbalance to the pressure to make educated guesses which often result in hallucinations [4].

*Self-consistency* [5] effectively mitigates a class of hallucinations by sampling multiple generations from an LLM in response to a given prompt and a final answer is selected by taking a majority vote over the responses. This approach, and follow up work [6] demonstrated improvement over alternative methods such as debate [7] and self-reflection [8], and that smaller models using this approach can match or surpass the accuracy of substantially stronger models. This method can be understood as mitigating a class of hallucinations where a model is able to produce correct answers in response to a given prompt more often than not, whether by means of imperfect recall or intelligent inference based on known/provided information.

---

*Corresponding author: `demian.till@cambridgeconsultants.com`

39th Conference on Neural Information Processing Systems (NeurIPS 2025) Workshop: Reliable ML from Unreliable Data.

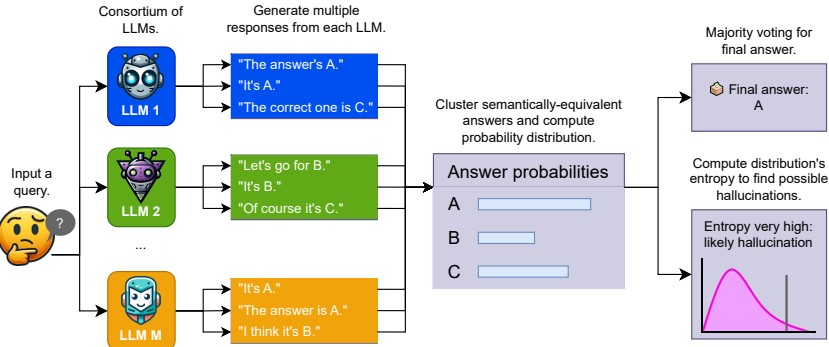

Figure 1: Illustration of consortium consistency. A given query is input to multiple LLMs, one or more responses are sampled from each model. Semantically-equivalent answers are clustered together, and the probability distribution of different answers is computed from these clustered samples. The distribution is used to calculate a final answer to the query, and an entropy score. Queries with higher entropy have less consistent responses and are hence more likely to contain hallucinations. Combining responses from multiple different LLMs reduces the likelihood of incorrectly assigning high confidence to hallucinated answers and allows consistently hallucinating models to be out-voted by other models.

*Semantic entropy* [9, 10] uses a similar consistency-based approach to detect likely hallucinations by grouping together similar generations sampled in response to a given prompt and computing the entropy over the resulting clusters of responses. An LLM is deemed more likely to be hallucinating when its responses to a given prompt have higher semantic entropy, reflecting greater uncertainty and more guesswork. The authors showed that semantic entropy achieves greater hallucination detection accuracy than alternative methods, including ones requiring white-box model access and model training [10]. [11] extend this idea to compute consistency across multiple responses based on semantic information within internal model embeddings rather than output text, achieving state-of-the-art (SOTA) hallucination detection results.

However these single-model consistency approaches naturally fail in cases where models produce relatively consistent hallucinations in response to a given prompt. In these cases the wrong answer can win the majority vote (hallucination mitigation failure) and semantic entropy can be low [9] or internal embeddings can be semantically similar [11], indicating that the answer is unlikely to be a hallucination (hallucination detection failure). We hypothesize that heterogeneous models with different training data, training methods and model architectures are less likely to share the same shortcomings in their training data or to making the same educated guesses. Heterogeneous collections of models therefore ought to be less prone to the aforementioned failure modes which arise when using single-model consistency approaches.

This motivates extending single-model consistency methods to incorporate multiple different LLMs. We therefore propose *consortium voting* and *consortium entropy* as multi-model counterparts to self-consistency (single-model voting) and semantic entropy respectively. These multi-model formulations, which we refer to collectively as *consortium consistency*, work in tandem to select answers and estimate confidence in selected answers from a pool of candidate responses generated by two or more LLMs. Other consistency-based hallucination detection methods such as [11–14] could similarly be extended to use multiple different LLMs, however we leave this to future work, and in the case of [11] this would require aligning the embedding spaces of different LLMs.

We compare consortium consistency with *single-model consistency*, which analogously uses self-consistency and semantic entropy in tandem with a single model. We evaluate both approaches on a set of 11 tasks, testing for reasoning capabilities, general knowledge, and domain-specific knowledge across a variety of domains. We explore consortia formed using various combinations from a pool of 15 different LLMs, ranging from 6B to 141B parameters in size, and using a range of different architectures, training methods, and training datasets.

We find that for many combinations of models, consortium voting and consortium entropy substantially outperform their single-model consistency counterparts whilst simultaneously reducing

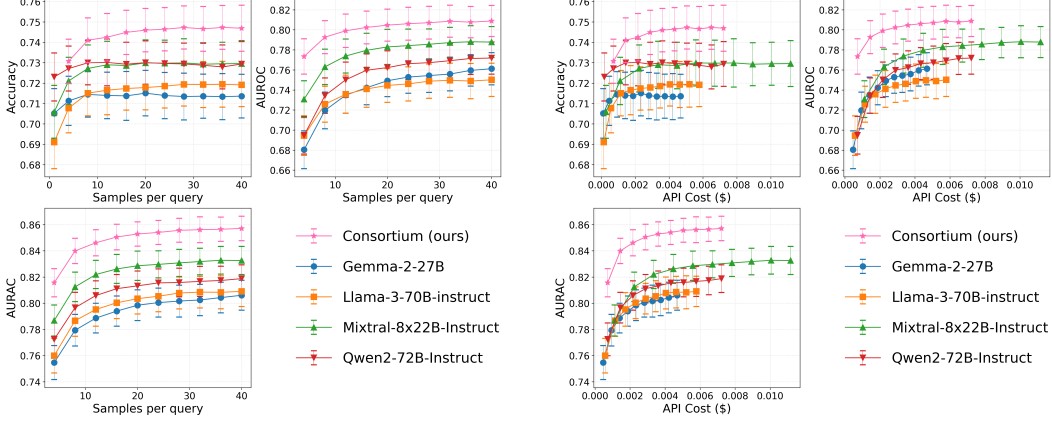

(a) Performance versus number of samples      (b) Performance versus API cost

Figure 2: (a) A representative example showing consortium consistency improving on average across 11 test sets over single-model consistency applied to each of the constituent models, across a range of sample budgets per-query. (b) Consortium consistency dominates single-model consistency on the cost-performance frontier, achieving both higher performance and lower cost simultaneously. X-axes show mean API cost in dollars per query, which grows with increasing number of sampled responses per query.

inference costs. However we also find that these performance gains are sensitive to consortium composition i.e. which LLMs are teamed together. We therefore investigate under what conditions consortium consistency tends to deliver the strongest results compared to single-model consistency.

In summary, our main contributions are[2]:

- We propose *consortium voting* and *consortium entropy*, collectively referred to as *consortium consistency*: black-box, post-training methods which further advance LLM hallucination mitigation and detection capabilities beyond their single-model consistency counterparts.

- We evaluate these methods using a wide variety of tasks, across a broad range of consortia composed of varying combinations of models from a pool of 15 diverse LLMs, finding that under reasonable constraints in consortium composition, consortium consistency outperforms single-model consistency when controlling for sample budget and model availability.

- We investigate which factors regarding consortium composition result in the most reliable improvements in performance compared to single-model consistency baselines, finding that performance gains tend to be greatest when all of the LLMs in a consortium are similarly capable and relatively strong (i.e. high-performing LLMs are better at complementing the capabilities of other high-performing LLMs).

- We additionally find that sometimes stronger models are able to benefit from being teamed with much weaker models, resulting in substantially reduced inference cost compared with single-model consistency, whilst simultaneously boosting hallucination detection and mitigation performance compared with single-model consistency with the entire response budget allocated to the stronger model.

## 2   Methodology

Our approach is illustrated in Figure 1. Given an input query $x$, a set of models $\mathcal{M} = \{m_1, m_2, \ldots, m_{|\mathcal{M}|}\}$, and a total sampling budget of $N$ responses, we begin by sampling $N/|\mathcal{M}|$ responses from each model i.e. evenly distributing our sample budget over the $M$ models. Each response is generated independently using nucleus (top-$p$) sampling with temperature scaling [15]. These responses are then clustered based on semantic equivalence as described below. We propose

---

[2]Code to follow.

two related methods: *consortium voting* and *consortium entropy*, for respectively generating a final answer and providing a confidence estimate in that answer being a hallucination. These methods are straightforward multi-model generalizations of the single-model consistency-based methods: *self-consistency* [5] and *semantic entropy* [9]. Since they work together in tandem to select and estimate confidence in answers, we refer to them collectively as *consortium consistency*, and we similarly refer to self-consistency and semantic entropy collectively as *single-model consistency*.

## 2.1 Semantic clustering of responses

Similar to [5, 9], consortium consistency requires first clustering the $N$ responses into a set of semantically distinct equivalence classes $\mathcal{C} = \{C_1, C_2, \ldots, C_{|\mathcal{C}|}\}$, where all responses within a cluster are considered equivalent in meaning and $|\mathcal{C}|$ is determined automatically by the clustering algorithm. For example given the prompt "What is the capital of France", the responses "Great, question! Paris is the capital of France", and "The capital of France is Paris" would be considered semantically equivalent for our purposes.

To determine equivalence of responses when clustering, we follow [6] in using task-specific approaches. For multiple-choice tasks, responses are deemed equivalent if they select the same final option, regardless of their reasoning paths. For math tasks, responses are deemed equivalent if their final answers are mathematically equivalent, again regardless of reasoning paths. We also use these equivalence checks when comparing final answers against ground truth answers during evaluation. More general equivalence checking is possible, e.g by prompting another LLM to determine equivalence as in [9], but for convenience we restrict our focus to domains where equivalence can be computed algorithmically.

## 2.2 Multi-model response generation via consortium voting

Given a set of clustered responses, consortium voting determines the final answer via majority voting. That is, it determines which cluster has the most responses across all $M$ models:

$$\text{answer} = \arg \max_{C_i \in \mathcal{C}} \sum_{m \in \mathcal{M}} \sum_{j=1}^{N/|\mathcal{M}|} \mathbf{1}[r_{m,j} \in C_i] \tag{1}$$

where $r_{m,j}$ denotes the $j$-th response sampled from model $m$, and $\mathbf{1}[\cdot]$ is the indicator function.

We compare this to single-model majority voting (referred to in the literature as *self-consistency* [5]), where all $N$ responses are drawn from a single model:

$$\text{answer}_{\text{single}} = \arg \max_{C_i \in \mathcal{C}} \sum_{j=1}^{N} \mathbf{1}[r_j \in C_i] \tag{2}$$

## 2.3 Hallucination detection via consortium entropy

To estimate hallucination likelihood, we extend semantic entropy [10] from single-model settings to multi-model consortia. For input query $x$, we estimate the consortium's distribution over equivalence classes f as:

$$P(C_i \mid x) = \frac{1}{N} \sum_{m \in \mathcal{M}} \sum_{j=1}^{N/|\mathcal{M}|} \mathbf{1}[r_{m,j} \in C_i] \tag{3}$$

Then the consortium entropy is the semantic entropy over the clustered responses from all models in a consortium:

$$\text{SE}(x) = - \sum_{C_i \in \mathcal{C}} P(C_i \mid x) \log P(C_i \mid x) \tag{4}$$

As in [9], the semantic entropy for a given input query reflects the level of diversity of responses across distinct semantic equivalence classes. A value of zero indicates unanimous agreement, while higher values indicate greater uncertainty and therefore a greater likelihood of hallucination. Unlike token-level entropy, which may overstate uncertainty due to superficial differences such as different

ways of phrasing equivalent responses, semantic entropy captures uncertainty at the level of response *meaning*.

We compare consortium entropy to single-model semantic entropy [9, 10] where all $N$ responses are drawn from a single model. In the single-model case the distribution over clusters simplifies to:

$$P_{\text{single}}(C_i \mid x) = \frac{1}{N} \sum_{j=1}^{N} \mathbf{1}[r_j \in C_i] \tag{5}$$

## 3 Experimental setup

### 3.1 Evaluation metrics

We report evaluation accuracy following [5, 6], as well as AUROC and AURAC following [9, 10]. Accuracy simply measures the percentage of evaluation inputs answered correctly. We use this as a proxy for hallucination *mitigation*, with higher accuracy generally indicating fewer hallucinations. For hallucination *detection*, we use AUROC to evaluate how well consortium entropy and single-model semantic entropy are able to distinguish correct from incorrect final answers, aggregating over all classification thresholds.

We also report *area under rejection accuracy curve* (AURAC), introduced in [10]. *Rejection accuracy* is the accuracy when only considering a subset of questions on which semantic entropy scores are above a given threshold. Less confident answers are considered potential hallucinations, and rejection accuracy effectively measures the resulting accuracy if the approach were to abstain from answering those questions. AURAC aggregates rejection accuracy across all confidence thresholds.

### 3.2 Baselines

Given the impressive results achieved by the single-model consistency methods that we extend, we use these single-model consistency methods as baselines. We evaluate consortium consistency on many different selections of $M$ models comprising different *consortia*. For each consortium we compare its performance on the metrics defined above with that of applying single-model consistency using individual models from the $M$ models in the consortium, controlling for sample budget.

Specifically, when we evaluate a given consortium of $M$ models, using a sample budget of $N$ responses per-question, we also evaluate the result of applying the single-model consistency methods to each of the $M$ models, in each case with the full sample budget of $N$ responses all allocated to that one model. We use these single-model consistency scores to define three baselines against which to compare the consortium score:

- **Hard baseline:** the highest of the $M$ single-model consistency scores on a given metric. This is the most difficult baseline to beat as it assumes that we know which of the $M$ models will perform best on the test data using single-model consistency (which often would not be known in practice).

- **Standard baseline:** the median of the $M$ single-model scores. This represents average performance of single-model consistency methods in the common case where we do not know *a priori* which of the $M$ models is best suited to the target domain.

- **Worst-case baseline:** the lowest of the $M$ single-model scores. This represents the worst-case performance of single-model consistency methods where the least suitable model for a given target domain is selected.

### 3.3 Sampling procedure

Unless otherwise specified, we generate $N = 40$ responses per input prompt, either (i) distributed evenly across the consortium of models $\mathcal{M}$, or (ii) drawn entirely from a single model (when evaluating single-model consistency). When $|\mathcal{M}|$ is not a factor of 40, we use the largest multiple of $|\mathcal{M}|$ less than 40 (e.g., $N = 39$ for $|\mathcal{M}| = 3$) and use the same $N$ for the single-model baselines to ensure fair comparison. All responses are sampled independently using nucleus sampling with top-$p = 0.9$ and temperature $= 0.5$, and chain-of-thought prompting [16], unless otherwise specified.

Table 1: Benefits of consortium consistency over single-model consistency baselines when composing consortia using well-matched, strong models. Results are averaged over the 586 consortia which met the following criteria: standard deviation of constituent mock benchmark scores $\leq 5$ and mean constituent mock benchmark score $\geq 70$. Each row reports either the mean percentage change in score vs the corresponding baseline ($\pm$ std) or the percentage of teams that outperform the corresponding baseline (i.e. where the change in score vs the baseline is positive).

| Metric | Baseline | Accuracy $\uparrow$ | AUROC $\uparrow$ | AURAC $\uparrow$ |
|---|---|---|---|---|
| | Hard | $+1.33 \pm 1.03$ | $+1.84 \pm 1.48$ | $+2.75 \pm 0.69$ |
| Mean score $\Delta$ (%) | Standard | $+3.70 \pm 1.20$ | $+5.63 \pm 1.46$ | $+5.39 \pm 1.09$ |
| | Worst-case | $+9.67 \pm 3.44$ | $+18.80 \pm 10.41$ | $+16.20 \pm 7.22$ |
| | Hard | 92 | 92 | 100 |
| % of teams improved | Standard | 99 | 100 | 100 |
| | Worst-case | 100 | 100 | 100 |

## 3.4 Uncertainty estimation

Unless otherwise specified, each evaluation metric for each model/consortium is computed using 100 bootstrap samples over the input questions and model/consortium responses. Reported values are mean averages across these samples. Where shown, error bars indicate one standard deviation across the bootstrap samples.

## 3.5 Models

We evaluate consortium consistency using varying subsets of models from a pool of 15 LLMs ranging in size from 6B to 141B parameters. These include models from the LLaMA, Mistral, Qwen, and Gemma families (full list in Appendix C). We experiment with different strategies for selecting which models to team together into consortia.

## 3.6 Datasets

We evaluate consortia and baselines on 11 tasks covering reasoning and general/domain knowledge:

- **Reasoning:** GSM8K [17] (200 randomly sampled questions), GPQA-Diamond [18].

- **General and domain knowledge:** 8 MMLU [19] subsets covering virology, world religions, jurisprudence, astronomy, public relations, anatomy, college chemistry, and global facts. TruthfulQA [2], which probes for common misconceptions.

We report metrics averaged across all 11 tasks to approximate performance in mixed-domain, real-world deployment settings.

## 3.7 Separate tasks for model selection

The strategies we propose for selecting which models to team together into consortia consider the relative and absolute capability levels of the candidate models. Ideally public benchmark scores would be used for these purposes, however we could not find any public benchmarks covering all 15 models. We therefore compiled a separate set of tasks with which to estimate a *mock benchmark score* for each model (detailed in Appendix D).

## 3.8 Compute costs

Gathering and processing all of the LLM responses discussed in this paper cost approximately $1000. The majority of this cost resulted from the API costs required for sampling 40 responses per question across the 11 main datasets and the 15 models we used.

# 4 Results

## 4.1 Performance with well-matched, strong models

Figure 2a shows a representative example of the benefits of consortium consistency over single-model consistency when applied to a set of similarly capable, relatively strong models. Consortium consistency outperforms single-model consistency applied to any of the individual constituent models across all three metrics and across a wide range of response budgets. Table 1 summarizes results aggregated across many such consortia, identified using the mock benchmark scores of their constituent models (detailed in Section D). Specifically we select teams with (i) a standard deviation in mock benchmark scores below 5 points and (ii) a mean mock benchmark score above 70%.

Across these consortia, consortium consistency delivers significant improvements over all single-model consistency baselines. Of particular note, consortium consistency outperforms the hard baseline in the vast majority of cases ($\geq 92\%$ of teams on each metric; see Table 1). The performance gap widens further against the other baselines, with over 99% of consortia outperforming the standard baseline on each metric, and all consortia outperforming the worst-case baselines.

## 4.2 Impact of model strength

Figure 3b shows how the advantage of consortium consistency over the hard baseline is impacted by the mean strength of the models within a consortium, as measured by their mean mock benchmark scores. We observe that as mean strength increases, the advantage of consortium consistency over the hard baselines grows more reliable across all three metrics. It was not immediately obvious to us why this should happen, since as mean strength increases, so do the baseline scores against which consortia are evaluated.

We hypothesize that this result is in part due to more capable models being more likely to make more intelligent (less random) guesses and mistakes, making them more likely to generate consistent (rather than random) hallucinations in response to some queries. Weaker models on the other hand tend to produce more varied responses when they hallucinate, corresponding to more random guesses. This can result in lower semantic entropies when stronger models hallucinate, making such hallucinations more difficult to detect using single-model semantic entropy. This leaves more room for consortium entropy to benefit from different models being less likely to all hallucinate in the same way, making low entropy on incorrect answers less likely.

Table 5 shows detailed results vs baselines for consortia selected based on high mean model strength. Compared with results from selecting based on both high mean model strength and low variance in model strength Section 4.1, AUROC scores are significantly lower, however 81% of these consortia still beat the hard baseline, with all of the 1580 consortia evaluated beating the standard and worst-case baselines across all three metrics.

## 4.3 Impact of variance in model capability

Figure 3a shows how the advantage of consortium consistency over the hard baseline is impacted by diversity in model capability within a consortium, as measured by the standard deviation of the constituent models' mock benchmark scores. We observe that as variance in model capability decreases, the advantage of consortium consistency over the hard baselines grows more reliable across all three metrics. This aligns with intuition: a relatively strong model is less likely to benefit by sharing its response sample budget with substantially weaker models than with other similarly capable models. However, interestingly we see in figure 3a that in the case of accuracy, many consortia with high diversity in model capabilities still exhibit substantial improvements over the hard baseline.

Table 4 shows detailed results for consortia selected based on low-variance in model capability only. Compared to selecting based on mean model capability (Section 4.2), AUROC improvements over the hard baseline are less reliable, with only 68% of consortia beating the hard AUROC baseline. We investigate this in Section A, finding that at the lower entropy regions, consortium consistency maintains a strong advantage over single-model consistency, however when using weaker models, this is often outweighed by consortia being more prone to producing higher entropies on correct responses due to a significantly higher chance of "dissenting opinions" when weaker models are used.

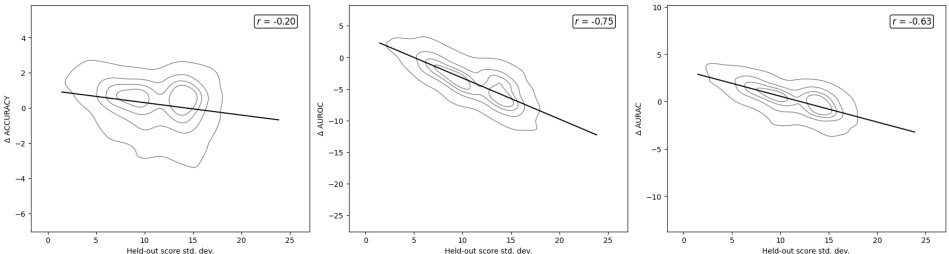

(a) Impact of diversity of mock benchmark scores on consortium performance vs hard baselines

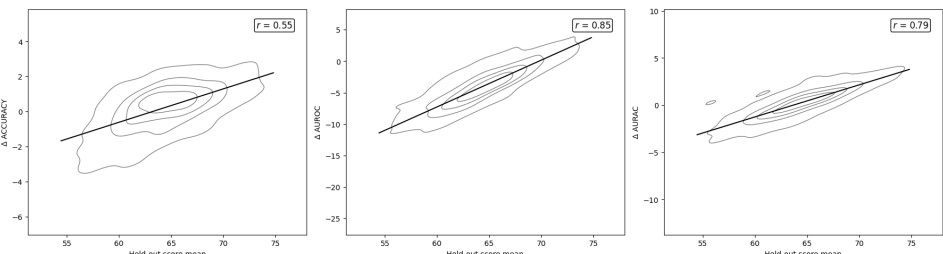

(b) Impact of mean individual model mock benchmark scores on consortium performance vs hard baselines

Figure 3: KDE plots showing performance of consortia vs hard baselines as a function of varying properties of constituent models. Plots are generated using 1000 consortia randomly selected from all $2^{15} - 15$ consortia that can be formed from the available models.

## 4.4 Cost-performance tradeoffs

Figure 2b compares performance against approximate API cost for consortium consistency and single-model consistency for a representative consortium and each of its constituent models. Across all evaluation metrics, the consortium dominates the single-model consistency baselines on the cost-performance frontier, achieving both higher performance and lower cost simultaneously. Note that this is an expected result because the strongest individual model is on average likely to be the most expensive. Therefore reallocating some of its sample budget across a consortium including cheaper models results in lower cost as well as the performance improvements which come with consortium consistency. See Section E for more examples of detailed plots for varied consortium compositions.

## 5 Related works

### 5.1 Hallucination detection

White-box methods have explored using token output probabilities [20–22] to calculate uncertainty scores, as well as training hallucination detection models using an LLM's internal embeddings [23, 24]. Black-box methods have explored prompting LLMs to provide confidence scores [25] and sampling multiple responses and evaluating consistency across responses [12, 13, 9, 10]. [26] uses a verifier model to check the answers of a target model, but is limited to teaming two models together in this manner. Other methods combine consistency across multiple samples with white-box model access [14], with [11] achieving SOTA results. Our method builds on [9, 10], inheriting the benefit of being black-box, and extends their approach to use an arbitrary number of different models, reducing the chance of unanimous agreement on hallucinated responses. However other consistency-based methods such as [11–14] could similarly be extended to use multiple models, and we believe they would likely see similar improvements as a result, but we leave this to be explored in future work.

## 5.2 Hallucination mitigation

While typically not explicitly framed as addressing hallucinations, certain consistency-based approaches [5, 27, 6] can be seen as mitigating a subset of hallucinations. Similarly, works combining the strengths of multiple models, both before generation of response(s) via model selection [28–31], during generation [32–34], and combining responses after generation [7, 35–37], can be understood as mitigating a different set of hallucinations arising from inherent limitations of individual models. Note that these works are concerned with improving the accuracy of generated answers rather than *detecting* hallucinations. Our work leverages the hallucination mitigation advantages of sampling multiple generations per-model and using multiple diverse models, whilst also estimating confidences in the selected responses which can be used for hallucination detection. Another line of work tackling hallucination mitigation involves retrieval-augmentation [38–40]. We see these approaches as largely orthogonal and potentially complementary to consistency-based approaches, and future work could look at combining them.

## 6 Conclusions and limitations

In this paper, we extended the single-model consistency methods: *self-consistency* and *semantic entropy*, proposing corresponding consortium consistency methods: *consortium voting* and *consortium entropy*. We demonstrated that consortium consistency improves over single-model consistency for mitigating and detecting hallucinations across a wide range of consortia, and does so whilst simultaneously reducing inference costs. We also analyzed the impacts of the capabilities of constituent models on consortium consistency performance relative to single-model consistency baselines, finding useful rules of thumb to help select models which are more likely to work well together in consortia. We hope that these results help motivate and guide future work in combining multiple LLMs for hallucination detection and mitigation.

Thus far, our evaluation has focused on average performance across 11 tasks. Further work could investigate how performance vs single-model consistency baselines varies with task diversity. Our hypothesis is that greater task diversity reduces the likelihood of any single model dominating, thereby enhancing the effectiveness of multi-model approaches. Conversely, in settings with more narrowly focused tasks, it may more often be preferable to rely on single-model consistency using the single strongest model in that domain when the best model is known.

While consistency-based approaches to hallucination detection and mitigation have achieved SOTA results, these come at the expense of substantially increased inference costs due to the need to sample multiple responses. This limits their applicability to situations where performance requirements outweighs inference cost concerns. We have seen that a multi-model approach can partially alleviate the increased costs due to the ability to combine models with varying inference-time demands, however this approach still incurs greater costs than more lightweight methods.

Recent work [37] indicates that stronger models can exhibit more similar failure modes, which could limit the benefits of multi-model consistency approaches. However, within the range of models explored thus far, we have observed the opposite, with stronger models benefiting more than weaker models from being teamed together. This could be the result of an interplay between two factors: the convergence of cross-model failure modes with increasing model strength indicated in [37] (which would harm consortium consistency), and the potentially even greater propensity for stronger single models to hallucinate more consistently (which would harm single-model consistency).

Another limitation of our approach arises when queries require knowledge of niche topics that few models within the consortium are experts in. Based on the current setup, a single expert model can be out-voted if some of the other models share a hallucination, perhaps based on some incorrect data points they share within their training data. To overcome this, further work may explore weighted aggregation based on known model strengths or per-model confidence estimates, to help ensure that authority can sometimes win out over consensus.

## Acknowledgments

We are grateful to James Oldfield, Douglas O'Rourke, David Rimmer, Rupert Thomas, and Joe Corrigan for valuable discussions and feedback.

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

# A  Precision-recall tradeoffs with weaker models

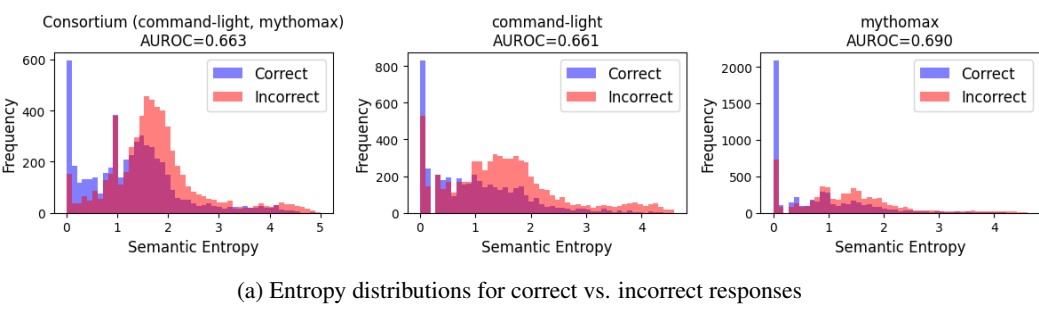

(a) Entropy distributions for correct vs. incorrect responses

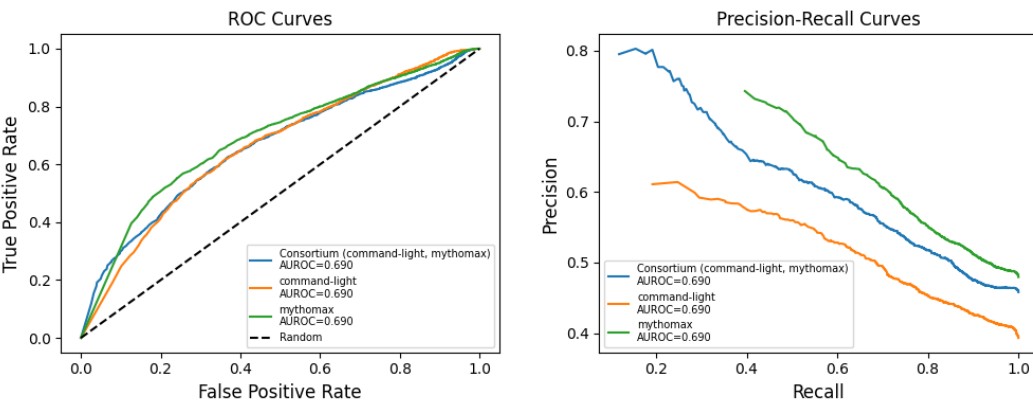

(b) ROC curves for individual models and the consortium

(c) Precision-recall curves showing trade-offs in precision vs. recall

Figure 4: (a) The entropy distributions show a clearer separation between correct and incorrect responses for the consortium in the low-entropy region, even though Mythomax achieves the highest AUROC overall. This suggests that the consortium is better calibrated in high-confidence cases. (b) While the consortium's improved low-entropy separation slightly boosts performance at the left-most part of the ROC curve, the overall AUROC remains highest for Mythomax. (c) Precision-recall curves reveal a more substantial benefit: while mythomax dominates in the higher recall range, the consortium attains higher peak precision, allowing more flexibility to trade off recall for higher precision. This is a common trend (see Appendix B).

Figure 4a shows entropy histograms for correct vs. incorrect responses in a representative consortium formed with weaker models. Consortium AUROC (0.663) is lower than that of the hard single-model consistency baseline, in this case provided by the Mythomax model, which has AUROC: 0.690. Despite the lower AUROC, the consortium has a significantly higher proportion of correct responses in the low-entropy region, indicating less propensity to hallucinate consistently than using single-model semantic entropy with Mythomax. Figure 4c shows the corresponding precision-recall curves: the consortium achieves substantially higher peak precision, although in this case at the cost of lower recall, with little impact on the ROC curve (Figure 4b). Appendix B presents additional examples for randomly selected consortia, showing that this is a typical pattern.

This supports our hypothesis that it is rare for multiple different models to hallucinate in exactly the same way, meaning that when near-unanimous agreement does occur, it is more trustworthy than in the single-model case. However it also highlights a limitation of consortium entropy for consortia formed with weaker models: consortia are more likely to have dissenting opinions even on correct answers, making it more difficult in some cases to distinguish hallucinations from non-hallucinations at the higher entropy range. This is particularly an issue for consortia with poorly (or in this case randomly) selected constituents. Recall that in Section 4.1 we showed that when selecting for consortia with low variance in strength and high mean strength of constituent models, overall AUROC scores are reliably improved over the hard baseline.

# B Precision-recall curves for randomly selected consortia

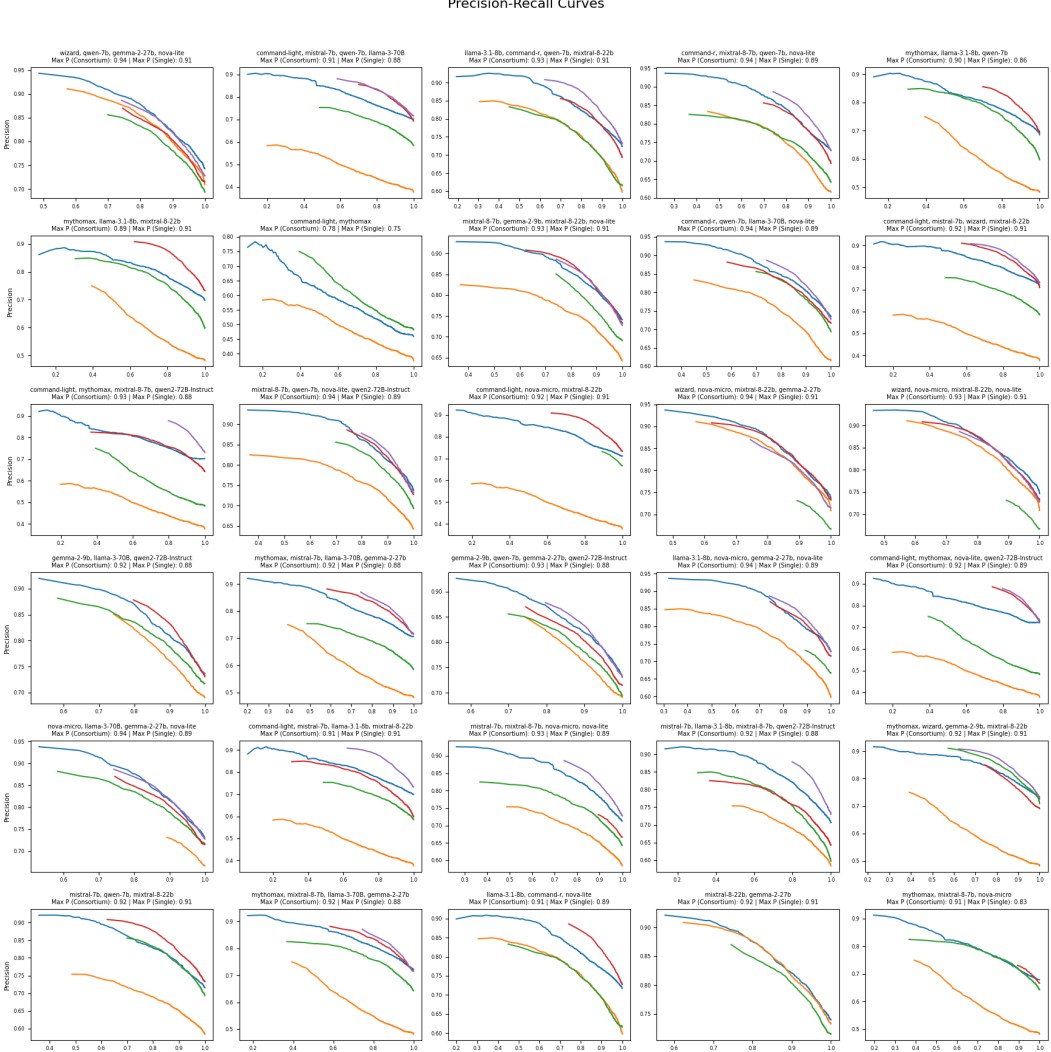

Figure 5: Consortium entropy typically attains higher peak precision values than semantic entropy with any of the individual constituent models (controlled for number of responses per query). Consortia typically allow more flexibility to trade-off recall for precision, even in cases where some of the individual constituent models have higher AUROC scores (see Section A for more discussion). Each sub-plot shows the precision-recall curve for a randomly chosen consortium (in blue) along with precision-recall curves for each of the constituent models. Consortia are chosen at random without filtering for variance in ability or mean ability, however they are filtered to consortia comprised of 4 models or less to aid readability.

# C  List of models used

Table 2: LLMs used in this study. Where available the model parameter count and API used for access is given.

| Abbreviated model name | Full model name | API | Model parameters [Billions] |
|:---:|:---:|:---:|:---:|
| mythomax | Gryphe/MythoMax-L2-13b-Lite | together.ai | 13 |
| nova-micro | amazon.nova-micro-v1:0 | AWS Bedrock | not published |
| nova-lite | amazon.nova-lite-v1:0 | AWS Bedrock | not published |
| llama-3.1-8b | meta.llama3-1-8b-instruct-v1:0 | AWS Bedrock | 8 |
| mistral-7b | mistralai/Mistral-7B-Instruct-v0.3 | together.ai | 7 |
| qwen-7b | Qwen/Qwen2.5-7B-Instruct-Turbo | together.ai | 7 |
| gemma-2-9b | google/gemma-2-9b-it | together.ai | 9 |
| gemma-2-27b | google/gemma-2-27b-it | together.ai | 27 |
| command-light | cohere.command-light-text-v14 | AWS Bedrock | 6 |
| command-r | cohere.command-r-v1:0 | AWS Bedrock | 35 |
| mixtral-8-7b | mistralai/Mixtral-8x7B-Instruct-v0.1 | together.ai | 46.7 |
| wizard | microsoft/WizardLM-2-8x22B | together.ai | 141 |
| mixtral-8-22b | mistralai/Mixtral-8x22B-Instruct-v0.1 | together.ai | 141 |
| llama-3-70B | meta.llama3-70b-instruct-v1:0 | AWS Bedrock | 70 |
| qwen2-72B-Instruct | Qwen/Qwen2-72B-Instruct | together.ai | 72 |

# D  Separate tasks for model selection

The strategies we propose for selecting model consortia which are likely to work well together consider the relative and absolute capability levels of the candidate models. For many models, public benchmark scores are available, however to our knowledge there are no public benchmarks on which all of the models in our experiments have been evaluated. Therefore, for the purposes of our experiments, we assembled a mock benchmark composed of a set of tasks covering a disjoint set of topics to those we use for evaluating consortia. Specifically, to compute a model's *mock benchmark score*, we evaluate it on 10 MMLU subsets not used in the main evaluation: abstract algebra, college computer science, college mathematics, econometrics, high school world history, human aging, marketing, philosophy, professional psychology, and sociology. Each model answers 100 questions per subset using greedy decoding, and we score them by their average accuracy (out of 100) across these tasks.

# E    Varied examples of consortia

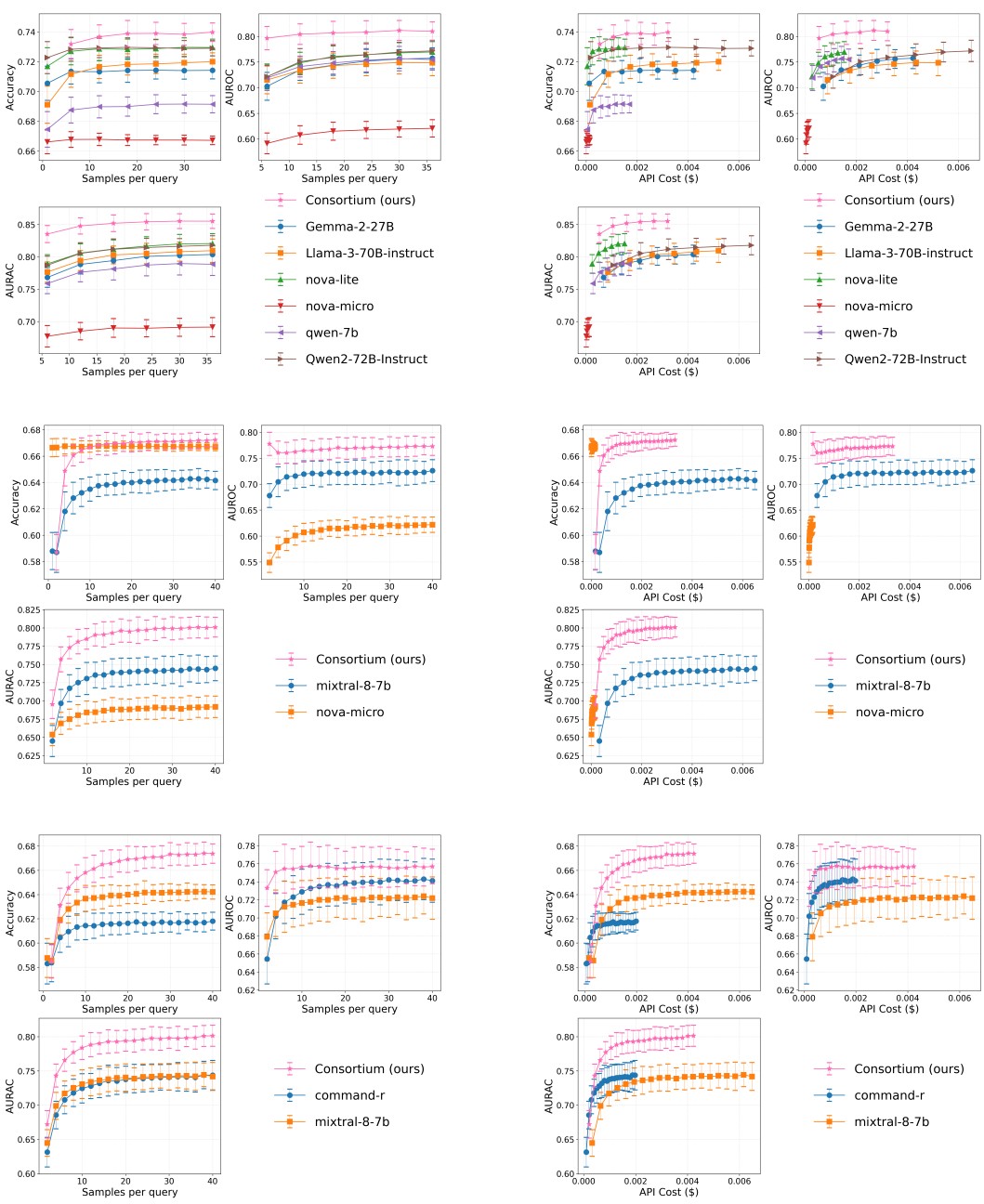

Figure 6: Varied examples of consortia which include weaker models and more variance in model capability outperforming single-model consistency applied to each of the constituent models.

# F    Detailed results

The following tables show the results of applying varying selection criteria when composing consortia.

Table 3: Benefits of consortium consistency over single-model consistency baselines when composing consortia using **well-matched, strong** models. Results are averaged over the 586 consortia which met the following criteria: standard deviation of constituent mock benchmark scores $\leq 5$ and mean constituent mock benchmark score $\geq 70$. Each row reports either the mean percentage change in score vs the corresponding baseline ($\pm$ std) or the percentage of teams that outperform the corresponding baseline (i.e. where the change in score vs the baseline is positive).

| Metric | Baseline | Accuracy ↑ | AUROC ↑ | AURAC ↑ |
|---|---|---|---|---|
| Mean score $\Delta$ (%) | Hard | $+1.33 \pm 1.03$ | $+1.84 \pm 1.48$ | $+2.75 \pm 0.69$ |
| | Standard | $+3.70 \pm 1.20$ | $+5.63 \pm 1.46$ | $+5.39 \pm 1.09$ |
| | Worst-case | $+9.67 \pm 3.44$ | $+18.80 \pm 10.41$ | $+16.20 \pm 7.22$ |
| % of teams improved | Hard | 92 | 92 | 100 |
| | Standard | 99 | 100 | 100 |
| | Worst-case | 100 | 100 | 100 |

Table 4: Benefits of consortium consistency over single-model consistency baselines when composing consortia using **well-matched** models. Results are averaged over the 928 consortia which met the following criteria: standard deviation of constituent mock benchmark scores $\leq 5$. Each row reports either the mean percentage change in score vs the corresponding baseline ($\pm$ std) or the percentage of teams that outperform the corresponding baseline (i.e. where the change in score vs the baseline is positive).

| Metric | Baseline | Accuracy ↑ | AUROC ↑ | AURAC ↑ |
|---|---|---|---|---|
| Mean score $\Delta$ (%) | Hard | $+1.24 \pm 1.14$ | $+0.87 \pm 2.03$ | $+2.32 \pm 1.23$ |
| | Standard | $+4.51 \pm 1.99$ | $+5.39 \pm 1.80$ | $+6.16 \pm 1.82$ |
| | Worst-case | $+11.93 \pm 4.70$ | $+19.35 \pm 10.14$ | $+17.00 \pm 6.49$ |
| % of teams improved | Hard | 90 | 68 | 98 |
| | Standard | 99 | 99 | 100 |
| | Worst-case | 100 | 100 | 100 |

Table 5: Benefits of consortium consistency over single-model consistency baselines when composing consortia using **strong** models. Results are averaged over the 1580 consortia which met the following criteria: mean constituent mock benchmark score $\geq 70$. Each row reports either the mean percentage change in score vs the corresponding baseline ($\pm$ std) or the percentage of teams that outperform the corresponding baseline (i.e. where the change in score vs the baseline is positive).

| Metric | Baseline | Accuracy ↑ | AUROC ↑ | AURAC ↑ |
|---|---|---|---|---|
| Mean score $\Delta$ (%) | Hard | $+1.43 \pm 0.88$ | $+1.17 \pm 1.42$ | $+2.58 \pm 0.64$ |
| | Standard | $+3.96 \pm 1.16$ | $+4.70 \pm 1.63$ | $+5.42 \pm 1.10$ |
| | Worst-case | $+16.76 \pm 6.89$ | $+17.87 \pm 10.09$ | $+18.01 \pm 6.05$ |
| % of teams improved | Hard | 95 | 81 | 100 |
| | Standard | 100 | 100 | 100 |
| | Worst-case | 100 | 100 | 100 |

Table 6: Benefits of consortium consistency over single-model consistency baselines when composing consortia randomly with **no filtering**. Results are averaged over 1000 random consortia. Each row reports either the mean percentage change in score vs the corresponding baseline (± std) or the percentage of teams that outperform the corresponding baseline (i.e. where the change in score vs the baseline is positive).

| Metric | Baseline | Accuracy ↑ | AUROC ↑ | AURAC ↑ |
|---|---|---|---|---|
| Mean score Δ (%) | Hard | +0.22 ± 1.18 | -4.03 ± 2.94 | +0.24 ± 1.46 |
| | Standard | +7.68 ± 3.55 | +1.34 ± 2.70 | +7.63 ± 2.82 |
| | Worst-case | +63.08 ± 29.58 | +16.98 ± 6.61 | +49.39 ± 22.27 |
| % of teams improved | Hard | 66 | 8 | 59 |
| | Standard | 100 | 72 | 100 |
| | Worst-case | 100 | 100 | 100 |

