# OpenReview forum: "Teaming LLMs to Detect and Mitigate Hallucinations"
_NeurIPS.cc/2025/Workshop/Reliable_ML — NeurIPS 2025 - Reliable ML Workshop_

### Official Review · Reviewer_YAJj · 2025-09-20
**Review for Teaming LLMs to Detect and Mitigate Hallucinations**

**Rating:** 6
**Confidence:** 4

**Review:**

### Summary
This paper proposes a method for hallucination detection and mitigation by combining outputs from multiple LLMs. It generalizes from self-consistency from single-model settings to multi-model ones. The authors introduce two techniques: consortium voting and consortium entropy. Using a pool of 15 models (param sizes ranging from 6B to 141B) across different families (LLaMA, Mistral, Qwen, …), they evaluate their LLM teaming approach over a variety of tasks. Their method consistently outperforms single-model consistency. Interesting, it also achieves improved cost-efficiency. The evaluation and experimentation are very solid, and the paper is structured/presented nicely so that it is easy for the readers to read.


### Strengths
- The authors have been sharing their thoughts/hypotheses throughout the paper, even on aspects that are not necessarily being the focus of the experiment. For example, I like their discussion in Section 6: “Our hypothesis is that greater task diversity reduces the likelihood of any single model dominating, thereby enhancing the effectiveness of multi-model approaches. […] is known.”
- Thorough empirical evaluation over diverse tasks and models, with well-designed baselines
- Figures and tables are clear, and the example shown in the figure is helpful for understanding their pipeline.

### Limitation & Suggestions for Authors
- I find the novelty of the paper to be incremental. In previous works such as “Improving Factuality and Reasoning in Language Models through Multiagent Debate”, they have tried using multiple LLMs from different families (e.g. Bard and ChatGPT) for debates / discussions to get a more accurate answer.
- Although the authors have included 15 models, there’s no inclusion of frontier models such as GPT-4o/5/o3 and Gemini 2.5 Pro — this limits the generality of their conclusion, particularly about real-world applicability. Given a fixed budget, it would be interesting to show whether it is better to use a consortium of weaker models like demonstrated in the paper, or is it better to use ChatGPT/Gemini directly? Or whether their finding about the benefits of teaming LLMs can transfer to the most powerful models in the industry, and these findings will more likely help engineers decide how to improve the performance of their application.
- Would be great if the authors can also experiment on whether keep adding more models into the team is always better. At what point do we stop getting significant benefits? What’s the trade-off between accuracy and compute budget in the context of multiple LLM teaming?